# The Role of the Cervicovaginal Microbiome on the Genesis and as a Biomarker of Premalignant Cervical Intraepithelial Neoplasia and Invasive Cervical Cancer

**DOI:** 10.3390/ijms21010222

**Published:** 2019-12-28

**Authors:** Gislaine Curty, Pedro S. de Carvalho, Marcelo A. Soares

**Affiliations:** Programa de Oncovirologia, Instituto Nacional de Câncer, Rio de Janeiro 20231-050, RJ, Brazil; gcf.science@gmail.com (G.C.); pedro42sw@gmail.com (P.S.d.C.)

**Keywords:** cervical microbiota, cervical intraepithelial neoplasia, cervical lesion, invasive cervical cancer

## Abstract

The microbiome is able to modulate immune responses, alter the physiology of the human organism, and increase the risk of viral infections and development of diseases such as cancer. In this review, we address changes in the cervical microbiota as potential biomarkers to identify the risk of cervical intraepithelial neoplasia (CIN) development and invasive cervical cancer in the context of human papillomavirus (HPV) infection. Current approaches for clinical diagnostics and the manipulation of microbiota with the use of probiotics and through microbiota transplantation are also discussed.

## 1. Introduction

Cervical cancer (CC) is the fourth most common cancer in women worldwide, with an incidence estimate of 569,847 cases and 311,365 deaths according to the latest Globocan report [1]. Cervical cancer is almost invariably caused by human papillomavirus (HPV) infection. There are currently over 200 different HPV types described that infect epithelial cells [2], of which around 40 have tropism to mucosal tissues. These are further divided into low-risk and high-risk HPV (lr-HPV and hr-HPV, respectively), depending on their carcinogenic potential [3]. lr-HPV are associated with the development of anogenital warts, while hr-HPV types are associated with cervical intraepithelial neoplasia (CIN) and CC [3]. Of the hr-HPV, HPV-16 and HPV-18 are responsible for approximately 70% of CC cases worldwide [4,5].

Although HPV infection per se is not sufficient to promote CC, the development of persistent infection is a major factor for cervical lesion progression and cancer outcome. The majority of HPV-infected women do not develop cervical cancer because immune response control infection, preventing cervical lesion development and its progression to cancer [6]. Thus, only a small fraction of infected women are not capable to control infection and develop CC. This fact suggests that additional factors might influence the progression of CIN to CC or its regression. The cervical microenvironment is complex, composed of immune cells and its specific microbiota that modulate local immune responses [7,8,9]. Recently, some studies have shown an association between the cervicovaginal microbiome and HPV infection, as well as CIN and CC [9,10,11,12,13,14]. In the current review, we describe the cervicovaginal microbiome landscape and its influence on the modulation of cervical viral infections, in addition to its role as a biomarker with predictive value for HPV infection and cervical neoplasia progression.

## 2. History of Bacteria Identification

The human body is a surprising ecosystem that presents trillions of microorganisms. The coevolution that occurred between man and microbes created a complex interaction network [15,16]. Most microorganisms are bacteria that reside in the gut and have a profound influence on the physiology and health of their hosts [17].

The identification of bacteria began in 1876 when Robert Koch created the methods to culture, isolate and identify the Anthrax agent [18]. After culture and isolation, bacteria were identified based on morphology, differential coloration, type of culture medium in which they grew, and biochemical characteristics. However, since the number of bacteria that can be cultivated and isolated is limited, such approach became insufficient to assess the complexity of microorganisms that exists in a single sample. In 1977, Carl R. Woese and George E. Fox published the first work utilizing the 16S rRNA gene to identify bacteria and showed that the gene could be used to identify microorganisms using molecular phylogeny [19]. This approach was revolutionary to biology and now this gene is broadly used in research that involves bacteria identification.

The 16S rRNA gene has approximately 1.5 kb and nine variable regions (V1 to V9) intercalated by conserved regions that allow PCR amplification of different bacterial orders using universal primers [20]. For a long time, PCR amplification of 16S gene was followed by cloning and further sequencing using the Sanger method. Although cloning and sequencing by Sanger provide information about bacterial composition, the generation of a large dataset is necessary to assess the high diversity of bacteria present in a sample.

The 21st century advanced technology brought the capability to generate large volumes of sequencing data faster and with high precision. Today it is possible to characterize the diversity of microorganisms present at different human anatomical sites [21,22]. The first large effort to characterize such diversity in the human body was the Human Microbiome Project (HMC) [21,23]. The HMC started in 2008 and analyzed samples collected from 242 healthy people that contributed greatly to the understanding of microbiome composition in different body parts of healthy subjects, including the vaginal microbiome [21]. Currently, second generation high-throughput sequencing (HTS) technologies, such as those provided by Illumina platforms, are being extensively applied to microbiome studies. More recently, third generation HTS technologies have also reached and contributed to the microbiome research field [24,25].

## 3. Microbiome and Cancer

The concept of the human microbiome was first suggested in 2001 for Joshua Lederberg when he coined the term “microbiome” to refer to the community of commensal, symbiotic, and pathogenic microorganisms, which share the same space with and form a complex interaction with specific human tissue compartments [21,26].

Changes in human microbiome homeostasis may impair the symbiotic relationship between host and microorganisms, promote physiological changes in the individual, and lead to the development of diseases such as cancer [27]. Recent studies show that the microbiota plays an important role on the development of different cancer types suggesting its involvement in various carcinogenic mechanisms [28,29]. Accordingly, the microbiota composition has been associated with breast cancer. A breast microbiome with differences in its composition between women with breast cancer and healthy individuals has been reported [30]. Moreover, a report compared four major breast cancer types with respect to their microbial signatures and described differences in their bacterial, viral, fungal, and parasitic compositions. For instance, two breast cancer subtypes (triple-negative and triple-positive) were remarkably distinct from each other, exhibiting very different microbial patterns [31]. Besides that, microbiome compositions were related with prostate health and disease. In fact, urinary, gastrointestinal, and oral microbiota signatures were associated with prostate alterations, including cancer [32]. Also, fecal microbiome analysis exhibited differences between prostate cancer and healthy samples [33]. Taken together, even though those results do not specify causality, they suggest a strong association between microbiome signatures and cancer at different anatomical sites.

The relationship between lung cancer and local bacterial composition has also been explored. The lower airways of lung cancer patients exhibited an enrichment of oral commensal bacteria (such as *Streptococcus* and *Veillonella*), which was related to upregulation of signaling pathways commonly associated with lung carcinogenesis [34]. In the gastrointestinal tract, the relationship between cancer and the microbiome composition has been extensively described. For example, a report compared the fecal microbiota from patients with neoplasia at different gastrointestinal sites (stomach, pancreas, small intestine, colon, and rectum). It was shown that the microbial patterns were different from control samples and, interestingly, the bacterial composition changed according to the affected site [35], suggesting that each gastrointestinal cancer has its own microbial signature. Concerning the esophagus, a shift in its microbiome to a less diverse community with dominance of single bacterial species (such as *Campylobacter* and *Lactobacillus*) was correlated with esophageal adenocarcinoma [36]. Further, *Helicobacter pylori* infection is a recognized risk factor for gastric cancer development, but other species from the gastric microbiome are also being associated with carcinogenesis [37]. In addition, intestine microbiome composition is linked with cancer development as a positive association between *F. nucleatum* and colorectal carcinoma has been reported [38,39]. Anal microbiome composition was also reported to change in Nigerian men who have sex with men (MSM) according to HIV risk, hr-HPV prevalence, sexual practices, and other factors [40].

Gynecological cancers have also been associated with microbiome constitution [41]. A study compared the microbiomes from different sites of the female reproductive tract (vaginal, uterine, Fallopian tubes, and ovarian samples) between women with endometrial cancer, endometrial hyperplasia, or benign uterine alterations. The authors described that endometrial cancer and hyperplasic patients had microbiome signatures distinguishable from that of the group with benign conditions, being *Atopobium vaginae* and *Porphyromonas* spp. particularly increased in the gynaecological tract from cancer subjects [42]. Of note, *Fusobacterium* spp. were also found in high prevalence among cervical cancer samples from Mexican women [9]. Also, microbiome analyses of ovarian cancer samples exhibited unique viral, bacterial, fungal and parasitic signatures that differ from the findings in healthy ovarian tissues [43]. Lastly, cervical neoplasia has been constantly associated with an increased diversity of vaginal microbiota [44], a relationship that will be further addressed in this review.

Altogether, the reports mentioned above harbor evidence from distinct organs linking the microbiome to cancer outcomes. However, the direction of this relationship is not completely understood. Whether microbes could act as carcinogenic agents leading to neoplasia or the tumor microenvironment modulates its surrounding microbial community still remains to be elucidated. Despite that, several microbial carcinogenesis mechanisms are being described, suggesting a driver role of microbes on cancer development (Figure 1). For example, microorganisms can produce toxins able to modify host cells and even modulate the immune system affecting its functionality [45,46]. In this context, some bacterial strains were reported to produce compounds referred to as genotoxins able to induce DNA damage [46]. As an example, specific *Escherichia coli* strains harboring the *pks* genomic island produce colibactin, a toxin that causes DNA double-strand breaks (DSB) in cultured mammalian cells [47]. Besides the DSB-induced damage response, colibactin-producing *E. coli* were also reported to modify the physiology of intestinal epithelial cell lines leading to an increased production of growth factors and, therefore, stimulating cell proliferation [48]. The cytolethal distending toxins (CDTs) are genotoxins commonly found in gram-negative pathogens and are composed of three subunits referred to as CdtA, CdtB, and CdtC. CDTs have tripartite structures and, while CdtA and CdtC are required to deliver properly CdtB into cells, CdtB induces DNA damage [49]. Indeed, CDT exposure of human colon epithelial cells lines increased genetic instability, especially in APC- or p53-deficient cells [50].

In addition to genotoxic effects, microbial communities have also been reported to modulate cancer through induction of an inflammatory milieu. In fact, microbial components induce activation of tumor-associated myeloid cells, leading to enhanced secretion of IL-23 and IL-17, cytokines able to stimulate colorectal tumor growth in mice [51]. Similarly, *E. coli* was shown to induce a pro-carcinogenic effect only when an inflammatory host response was present [52] and colorectal tumors with high abundance of *Fusobacterium nucleatum* exhibited a proinflammatory expression pattern [53]. Taken together, although all of those previous reports were described for colorectal cancer (CRC), they strongly suggest inflammation-associated carcinogenesis as a mechanism by which microbiota affect the immune system and participate in neoplasia development. Accordingly, mice with toll-like receptor 4 (TLR4, a receptor for bacterial lipopolysaccharide) constitutively activated registered colitis-associated neoplasia more often than WT counterparts [54], while TLR4 knockout animals developed significantly less colonic tumors when compared to controls [55]. However, it is noteworthy that bacteria can also signal directly to other cells independently from the immune system. For example, the adhesin FadA expressed by *F. nucleatum* binds to E-cadherin expressed by CRC cells, inducing signaling pathways that lead to proliferation and other oncogenic responses [56], therefore participating in carcinogenesis without necessarily eliciting inflammation.

Several studies have reported a microbial-dependent immune evasion that could facilitate tumor establishment. The γ-glutamyl transpeptidase (GGT) protein produced by *H. pylori* was able to abrogate T lymphocyte proliferation and induced its cell cycle arrest in G1 phase [57], while *F. nucleatum* density in CRC tissues was inversely proportional to T-cell abundance [58]. Those are suggestive evidence that microbes can act by hindering immune responses, creating a mechanism that favors tumor cell survival. In this context, the *F. nucleatum* Fap2 protein was reported to interact with an inhibitory receptor of NK cells (TIGIT), leading to cytotoxic activity blockade and tumor survival [59]. Also, gastric epithelial cells incubated with *H. pylori* expressed B7-H1 (PD-L1) and induced naïve T-cells to differentiate into a regulatory T-cell (Treg) phenotype, an immunosuppressive pathway [60]. Those results suggest that the bacterial composition within the tumor microenvironment can directly drive immune escape. Even the efficacy of anticancer immunotherapy using antibodies that target and block CTLA-4 (an inhibitory immunoreceptor) was demonstrated to change according to the gut microbiota constitution [61]. Therefore, microbiome composition is being currently proposed as a predictive marker for immunotherapy success [62,63].

## 4. Cervicovaginal Microbiome

### 4.1. Cervicovaginal Bacterial Composition and Profiles

The cervicovaginal microbiota can be classified into five groups based on the 16S rRNA high-throughput sequencing (16S-HTS) data according to the bacterial species present [64]. Those groups are called community state types (CSTs), a term that was first attributed by Ravel in 2011 [64]. CSTs were named from I to V according to dominant bacteria. CSTs I, II, III, and V contain *Lactobacillus crispatus*, *L. gasseri*, *L. iners,* and *L. jensenii* as the dominant species, respectively. CST IV exhibit bacterial high diversity, with increased frequency of anaerobic species such as *Gardnerella, Megasphera, Atopobium,* and *Prevotella* [64,65,66]. The cervicovaginal microbiota unbalanced composition, consisting of high diversity and low *Lactobacillus* abundance, like CST IV, characterizes a state named dysbiosis. Some women with dysbiosis develop symptoms such as abnormal vaginal discharge, inflammation, odor, and pruritus, being diagnosed, under these conditions, with bacterial vaginosis [67,68]. Although some women show symptoms, a great part of them are asymptomatic [65]. However, both symptomatic and asymptomatic women are more likely to acquire HIV, HPV, and other infections [66].

In general, the *Lactobacillus* genus shows high abundance in cervicovaginal microbiota and was first described in 1892 by Döderlein [69]. *Lactobacillus* species like *L. crispatus, L. gasseri,* and *L. jensenii* are able to produce lactic acid and hydrogen peroxide (H_2_O_2_), which inhibit the growth of other bacteria and viruses [70]. On the other hand, *L. iners* is considered a transitional species to the dysbiosis state [71].

The composition of the cervicovaginal microbiota is dynamic, changing due to the hormonal fluctuations that occur during women’s reproductive cycle, use of oral contraceptives, sexual activity, vaginal douching, lactation, *diabetes mellitus,* and stress [72]. During puberty, there is an increase in estrogen promoting the maturation, proliferation, and accumulation of glycogen in the vaginal epithelium [72]. Glycogen is catabolized to smaller polymers by the alpha-amylase present in the vaginal epithelium, which is further metabolized to lactic acid by *Lactobacillus* species [72]. Of the two acid lactic isomers (D- and L-lactic acid), D-lactic acid is more protective against infections [72,73]. Interestingly, differently from other *Lactobacillus* species, *L. iners* has a small genome and is unable to produce D-lactic acid and H_2_O_2_ [71,72]. Instead, *L. iners* produces L-lactic acid and several studies correlate the presence of the latter with viral infections. *L. iners* can produce inerolysin, a cholesterol-dependent cytolysin [74]. Inerolysin is a pore-forming toxin like the vaginolysin protein secreted by *Gardnerella* [71,74,75] which forms pores in vaginal epithelium, compromising its integrity and favoring viral infections. Besides lactic acid production, *Lactobacillus* species also produce antimicrobial peptides such as bacteriocins and biosurfactants, which inhibit pathogen growth and establishment [11,72].

### 4.2. Cervicovaginal Microbiota and Relationship to Viral Infection

The microbiota plays an important role controlling viral infections, such as those caused by HPV or HIV [10,76,77,78,79]. Viral infections are responsible for approximately 15% of cancer cases worldwide and viruses that are associated with cancer include HIV, Epstein-Barr virus (EBV), Kaposi’s sarcoma herpesvirus (KSHV) or human herpesvirus type 8 (HHV-8), human papillomavirus (HPV), and hepatitis B and C viruses [80]. Those viruses use different strategies that lead to carcinogenesis either affecting directly the cellular machinery or through of an indirect mechanism, by immune response inhibition or chronic inflammation. On the other hand, viruses may not be the sole determinants of certain malignancies. HPV infection, for example, is a necessary but not sufficient cause for cervical cancer development. It is not known why most women infected by HPV eliminate the infection and do not develop cervical cancer. Interestingly, the microbiota is the first line of contact against infections, and depending on its composition is able to produce lactic acid and H_2_O_2_ that have a protective effect against viral and bacterial infections [11,72] (Figure 2). Cross-sectional studies demonstrate a negative relationship between HPV infection and CIN with *Lactobacillus* dominance, except for *L. iners*. The *L. iners* microbiota (CST III) has also been associated with higher frequency of HIV, HPV, and HSV-2 infection [71,81]. A longitudinal study comparing the microbiota of HPV-positive and negative women also demonstrated that the presence of *L. gasseri* is positively associated with HPV elimination [79].

Although there are several studies on the cervicovaginal microbiome and its association with viral infections, little is known about the role of microbiome on carcinogenesis and on the mechanisms responsible for HPV elimination or persistent infection. Recently, a general model of virus–bacteria–host interaction highlighting two distinct mechanisms was proposed for the contribution of microbiota to virus-associated cancers [76]. The first one suggests that the microbiota affects directly viral infectivity through generation of bioproducts that could be able to module virus-host interactions. The second proposes that bacteria-host interactions affect the host gene expression and this modulation on its turn affects viral production and could promote the tumorigenesis associated with viral infection. It is important to highlight that the microbiota is also able to protect the host from viral infections and these mechanisms vary depending on the microbiota composition as discussed above.

The presence of CST IV in the cervicovaginal niche is associated with a higher risk of developing HPV persistent infections and consequently cervical lesions [10,11]. *Gardnerella vaginallis*, present in CST IV, is able to secrete sialidase that degrades the vaginal mucus by cleaving its glycoproteins [72,82]. The mucus contain proteins such as mucin that provide a physical barrier to the vaginal mucosal surface and inhibit bacteria-host interactions [11,72], and its degradation compromises the mucosal barrier and favor genital tract infections. Moreover, the cervicovaginal microbiota metabolism can also influence virus infectivity. Women that carry *Gardnerella*-dominant microbiota metabolize tenofovir administered intravaginally as a microbicide, which reduces the antiretroviral activity of the drug and increases risk of HIV infection acquisition [83]. Further, bacteria present in CST IV are also able to produce butyric acid that is able to regulate histone acetylation and many studies have shown the epigenetic regulation of that metabolite in the reactivation of latent HIV-1 proviruses, indicating a potential involvement of microbiota in AIDS progression [84].

The profile of CSTs varies according to women’s ancestry [64,65]. While women of European ancestry show a high frequency of CST I, women of Asian and African ancestries show high frequency of CST III and IV, respectively [64]. As discussed above, CST IV is associated with the acquisition of some viral infections, like HPV and HIV [8,11]. Recent studies show that women with the CST IV have high production of proinflammatory cytokines, which increases the recruitment of activated CD4+CCR5+ cells to the vaginal mucosa, favoring the acquisition of HIV [8]. Furthermore, the stimulation of cytokine production by the microbiota (such as TNF-alpha) damages the epithelial barrier, which could promote HPV infection, and this has been shown in women with CST IV [9]. Thus, women with dysbiosis can develop chronic inflammation, being this an important factor for cancer development in different tissue types, including cervical cancer [85]. On the other hand, the lactic acid produced by *Lactobacillus* is also able to modulate immune responses and influence viral infections. In vitro, lactic acid can act directly on the cervicovaginal epithelium, inducing the production of the anti-inflammatory cytokine IL-1Ra and reducing pro-inflammatory cytokine production. This has been observed in three distinct in vitro cellular models, cervicovaginal epithelial cell lines, primary vaginal epithelial cells and in organotypic 3D tissue (EpiVaginal tissue model) [86]. The ability of lactic acid to inhibit pro-inflammatory cytokine production induced by pathogen-associated molecular patterns can affect virus infection, such as that by HIV, since some immune mediators recruit and activate HIV target cells, favoring infection.

Proteomic studies have been performed on cervicovaginal fluid to analyze and understand the role of the microbiota on the cervical region metabolism [87,88,89]. A study using proteomic data showed that dysbiosis causes cervicovaginal inflammation and detrimental changes within the mucosal barrier [90]. Another recent study shows that cervicovaginal microbiome proteomic analysis may be conducted using the residual Papanicolaou test supernatants for community composition and functional microbiota characterization [87]. Proteomic analysis can be used in research to understand the cervicovaginal microbiome and its contribution to women’s health and disease in the metabolic context.

## 5. Microbiota as a Biomarker for HPV and Cervical Dysplasia

Although HPV infection is a necessary cause, it is not determinant for cervical cancer development. HPV infects squamous epithelial basal cells, inducing lesions and even cervical cancer when it is not eliminated [91]. However, the majority of HPV infections are cleared and only a small fraction of infected women progress to premalignant lesions and cancer [91]. Pap smear has been used for cervical screening, which resulted in a decrease in deaths from cervical cancer. Nevertheless, the assay has low sensitivity (60–80%), high false-negative rates (30%), and significant false-positive rates, ranging from 15–50% [92,93]. On the other hand, introduction of HPV-DNA assays for screening has improved the results from equivocal cytology triage with Pap smear [94].

In many countries, Pap cytology is the primary screening test either alone or in conjunction with HPV DNA test (co-testing), although in some European countries a switch to primary HPV-DNA technique followed by cytology (Pap smear) has been recommended [95]. The American Cancer Society recommends the use of the HPV test as part of follow-up for an abnormal Pap result in women aged 21–29, while for women aged >31 co-testing screening is recommended every five years [96].

The HPV-DNA test is a high-sensitive method, so the absence of high-risk HPV-DNA indicates low risk for CIN3 and cancer development, which may allow safe prolonging of cervicovaginal screening test intervals [95]. Additionally, even when HPV-DNA scores positive, the majority of HPV infections are eliminated and do not progress to cervical dysplasia. Nevertheless, since the risk of cancer development still exists, screening at short intervals is strongly recommended. Therefore, the characterization of novel biomarkers is important in order to decide precisely how each HPV-positive women will be treated (colposcopy) and whether HPV-DNA-negative women have high risk to acquire a new HPV infection and progress into CIN. They may function as secondary markers after HPV-DNA test and Pap smear to identify women under risk to HPV acquisition, persistent infection and cervical cancer, contributing to better follow-up/treatment strategies.

Acknowledging a biomarker potential, high diversity microbiota has been frequently shown to correlate with HPV status and different severities of cervical dysplasia, suggesting a potential in indicating vaginal health and disease (Figure 3). Therefore, we further discuss the cervicovaginal microbiome as a promising biomarker not only for HPV status but also for cytologic abnormalities.

Different reports consistently demonstrated the relationship between HPV infection and/or persistence and microbiome composition. In fact, a study reported that HPV+ women exhibited a more complex microbiome diversity than HPV- counterparts [97]. Similarly, a work with Korean twins reported that HPV+ patients had a high-diversity vaginal microbiome with reduced proportion of *Lactobacillus* spp. when compared to their HPV- matches. Moreover, bacteria of the *Sneathia* genus was remarkably associated with HPV positivity and suggested as a biomarker for viral status [78]. Taken together, those reports strongly suggest a relationship between microbiome increased diversity and HPV infection. On the other hand, they do not demonstrate a causality link between a shift in microbiota composition and HPV acquisition, since those studies were not longitudinal with a patient’s follow-up through time.

In this scenario, a study following women for 16 weeks showed that a vaginal microbiota dominated by *L. gasseri* was related to a faster HPV clearance, while microbiomes with lower *Lactobacillus* levels and higher abundance of *Atopobium*, *Gardnerella,* and *Prevotella* were associated with a slower infection resolution [79]. Further, *L. crispatus* was the most prevalent *Lactobacillus* spp. in Italian women that cleared HPV infection or were consistently HPV-, while a microbiome composition characterized for lower *Lactobacillus* spp. counts and higher abundance of *Gardnerella*, *Prevotella*, *Atopobium,* and *Sneathia* (a combination commonly associated with bacterial vaginosis) was the most frequent among women with persistent hr-HPV infection during one year. Also, among patients with persistent HPV infection, *L. iners* was the most abundant *Lactobacillus* species [98]. Accordingly, a longitudinal observation of African/Caribbean women living in Canada showed that a microbiome with lower abundance of *Lactobacillus* spp. and greater representation of anaerobic bacteria was more frequent in HPV+ than in HPV- subjects [99]. Altogether, those studies are important to elucidate the cause-effect relationship between vaginal microbiota and HPV status. Since they report a temporal dynamics in which infection resolution and persistence are associated with different bacterial compositions, microbiome alteration is suggested as a factor that could occur before HPV acquisition, modulating, and facilitating viral maintenance. Moreover, a possible protective role for *Lactobacillus* spp. dominated microbiota is noteworthy, while its paucity and increased amounts of other bacterial genera is related with higher HPV risk. Nevertheless, a study has recently provided some insights into possible interactions between viruses and the vaginal microbiome. The authors demonstrated that cervicovaginal samples with CSTs I or IV that were positive for oncogenic viruses (HPV and/or polyomaviruses) showed increased abundance of *L. crispatus* as well as *P. timonensis* and *S. sanguinegens*, respectively, when compared to its counterparts without any virus detected [100]. That suggested that the presence of viruses may also exert influence in the cervicovaginal microbiome composition. Ethnicity is another important factor that must be considered, since *L. gasseri* (together with *G. vaginalis*) was more frequent in HPV+ subjects compared to HPV- in a Chinese cohort [97] as well as *L. mucosae* and *Enterococcus faecalis* were the most dominant species among women from Northeast India [101], which opposes previous findings.

It is also important to highlight that microbiome composition has also been related to other viral infections. For example, African women with *L. crispatus*-dominated microbiota registered significantly less frequent HIV, HSV-2, and HPV infections and bacterial STIs when compared to other compositions [81]. A report studying Caucasian Italian women demonstrated higher rates of HPV infection among samples belonging to CSTs III or IV, as well as a higher frequency of polyomaviruses in women with CSTs III or I. Although the *L. crispatus*-dominated environment has been associated with the presence of polyomaviruses, a longitudinal analysis of those patients revealed that CST I was associated with increased rates of viral clearance, while women harboring CSTs III or IV commonly progressed to persistent infections [100].

As discussed above, the vaginal microbiota composition has been extensively associated with HPV positivity and suggested as a promising biomarker for HPV risk. Accordingly, HPV infection is strongly recognized as a necessary, but not sufficient, cause for cervical carcinogenesis. Therefore, since specific bacterial compositions are linked with increased viral infections, the microbial community could also be associated with cervical dysplasia development and inform about cytological abnormalities. Indeed, different studies have already described a cervicovaginal microbiome shift with increased proportions of bacteria such as *Gardnerella, Prevotella, Atopobium,* and decreased abundance of *Lactobacillus* spp. occurring together with CIN and cancer. However, again, the cause–effect relationship between microbiome and cervical dysplasia has not been elucidated well [11].

Differences in microbiota composition were found between normal cytology, cervical lesions and cancer. That is, while *L. crispatus* and *L. iners* were respectively the predominant species for HPV- and HPV+ women without cytologic alterations, *Sneathia* spp. and *Fusobacterium* spp. were predominant in squamous intraepithelial lesions and cervical cancer, respectively [9]. Similarly, CST IV (high diversity microbiome lacking *Lactobacillus* spp.) frequency was shown to be directly proportional to cervical abnormalities severity: the cluster was gradually more abundant in low-grade squamous intraepithelial lesions (LSIL), high-grade SIL (HSIL) and cervical cancer. Moreover, HSIL samples had greater abundance of *Sneathia sanguinegens*, *Anaerococcus tetradius,* and *Peptostreptococcus anaerobius* than LSIL, suggesting shifts in microbiome composition according to disease severity [13]. Likewise, *Lactobacillus* dominance decreased together with cervical dysplasia severity while *Sneathia* spp. were increased in low/ high grade precancerous lesions and invasive cervical carcinoma [102]. In this context, different analyses suggest a paucity of *Lactobacillus* and increased relative abundance of other species as a factor associated with cervical abnormality development. Those reports also indicate that each stage of cervical dysplasia is associated with a corresponding microbial pattern. For example, while *Lactobacillus* spp. are more associated with normal cytology and absence of HPV, a higher abundance of *Sneathia* spp. was observed throughout cancer development and, therefore, could be used as indicative of vaginal disease together with other species mentioned above.

Additional evidence corroborates the findings described above. First, the presence of *A. vaginae, G. vaginalis*, and *L. iners* together with *L. crispatus* in low levels was suggested as the most hazardous combination for CIN development, with an odds ratio of 34.1 for CIN in the presence of hr-HPV [14]. Moreover, a report indicated that *B. fragilis*, *L. delbrueckii,* and *S. agalactiae* had an indirect effect on cervical cancer mediated by HPV infection, while *A. vaginae* and *P. stutzeri* also exhibited a direct effect on cervical carcinogenesis independently of HPV status [103]. Of note, a study evaluated the impact of loop electrosurgical excision procedure (LEEP), a method to treat CIN 2/3 to avoid cancer development, in the microbiome composition. The authors noticed that a microbiota containing *Prevotella* and lacking a consistent dominant species shifted significantly to an *L. iners* dominated community after three months of LEEP intervention [104]. However, in sharp contrast to those findings, it has been reported that a microbiome dominated by unclassified *Lactobacillus* spp. and *L. iners* was significantly associated with CIN 2 and CIN 3 in women with hr-HPV [105]. Although *L. iners* was already described to be related with CIN [14] or HPV positivity [9], the dominance by other *Lactobacillus* species was previously reported as a protective factor, but this cohort showed a different observation for unclassified *Lactobacillus* spp. In this scenario, Hispanic ethnicity by itself was associated with a decrease in *Lactobacillus* dominance and *Sneathia* spp. enrichment when compared to non-Hispanic women living in the U.S [102].

HIV-positive women show increased risk of HPV acquisition and CIN development [106,107,108]. A study that analyzed the cervicovaginal microbiome in the postpartum period of HIV-positive women showed a high frequency of *L. iners, Moryella, Schlegelella,* and *Gardnerella* associated with CIN with significant odds ratios of 40 for *Moryella* and of 3.5 for *Schlegelella* [12]. In a longitudinal analysis, when comparing the bacterial microbiome of women that showed CIN regression to normal cytology, *Gardnerella* appeared with a higher frequency in CIN when compared to normal status [12]. Similarly, a report from HIV+ pregnant women in Zambia showed that they had higher microbiome diversity, greater abundance of *G. vaginalis* and *A. vaginae* and lack of *L. crispatus* when compared to HIV- pregnant participants. Also, *L. iners* enrichment was observed in HIV- individuals and in HIV+ subjects with preconceptional ART exposure [109]. These findings corroborate the association of specific bacteria with cervical premalignant lesions. Despite such association, it is important to highlight that little is known about the longitudinal nature of changes in microbiome in the development of CIN and cervical cancer in HPV+ women. However, the association seen in cross-sectional and in some longitudinal studies indicates that the microbiome could be used as a sensor for cervical alteration and risk for CIN progression.

## 6. Clinical Molecular Diagnostics

The bacterial abundance in the cervicovaginal microenvironment is variable and the term CST is used to indicate the dominant species present, as described above. CST IV is the most diverse type and has bacterial species linked with CIN, such as *Gardnerella*, *Prevotella,* and *Atopobium* as dominant species, and other diverse species with variable abundance, like *Anaerococcus sp., Peptostreptococcus sp., Fusobacterium sp., Moryella, Sneathia,* and *Schlegelella* [11,12,13]. The current approach to detect bacterial species in scientific studies is the 16S-HTS. This technique is able to efficiently detect the complexity of bacteria present in a unique sample and recent studies have discussed its use as clinical diagnostics [110].

Clinical samples encompass a complex array of multiple bacterial species, posing a challenge to proper microbial identification methods. The cost of 16S-HTS has decreased during the last decade and can be applied as a diagnostic method in medical microbiology laboratories [110]. The main challenge of its introduction as a diagnostic method is data analysis. However, the recent development of user-friendly pipelines and softwares for 16S analysis bypassed this obstacle [111,112,113,114,115]. On the other hand, alternative molecular diagnostic tests such as direct probe assays and real-time PCR have also been developed and commercialized to identify specific bacteria in the cervicovaginal microenvironment using smear samples.

Women with bacterial vaginosis (BV), irrespective of their symptomatic status, show higher frequency of viral infections and cervical lesions [11,66]. In general, diagnosis is done by Amsel criteria and direct Gram staining of vaginal secretions (Nugent score) [116,117]. The Amsel criteria is based on observation of at least three out of four parameters: clue cells on microscopy, thin watery homogeneous discharge, pH > 4.5, and fishy odor upon addition of 10% potassium hydroxide to the secretion. The Nugent score varies from 0–3 (normal), 4–6 (intermediate), to 7–10 (BV) and the values are attributed according to the number of large gram-positive rods (*Lactobacillus* morphotypes), gram-negative or gram-variable (BV flora) bacteria observed from the secretion collected through a Pap exam. Due to their limitations, novel methods using molecular diagnostics have been developed [118,119,120]. Commercial tests using DNA probes are able to detect *G. vaginalis* with high sensitivity (90–94%) and specificity (97–81%) when compared to Amsel criteria and Nugent score methods [120]. Further, there are some commercial real-time PCR tests that can also be used in BV diagnosis. The tests are based in semi-quantitative or quantitative multiplex real-time PCR assays and are able to identify different bacterial species, such as *A. vaginae*, BVAB-2, *Megasphaera* type 1 and 2, *L. acidophilus*, *L. crispatus*, *L. jensenii,* and *G. vaginalis* with high sensibility (90–99%) and variable specificity (70.2–95%) when compared to Nugent score and Amsel criteria [120,121]. The variable specificity can be explained by the high occurrence of asymptomatic vaginal dysbiosis [121]. Some bacteria measured in these tests are associated with HPV infection, persistence, cervical lesions, and cancer [11]. Therefore, all these methods, including 16S-HTS, can be implemented to analyze HPV+ women with potential risk to develop cervical lesions or viral persistence. In fact, the use of molecular techniques in clinical diagnostics has evidenced a wide range of suboptimal vaginal microbiome compositions. That is, the entities that were referred to as BV after Amsel or Nugent diagnosis (Amsel or Nugent-BV) represent only a small fraction of the altered microbiome compositions detected by molecular methods (Molecular-BV). In this scenario, the molecular detection of BV is clinically relevant since it is associated with unfavorable outcomes such as increased HIV acquisition risk and is also informative for patients whose suboptimal bacterial flora could not be detected by Amsel criteria and/or Nugent score [122].

## 7. Probiotics and Microbiota Manipulation

Women with BV show a high diversity microbiota and the most traditional treatment is the use of prescribed antibiotics such as metronidazole and clindamycin that do not ensure cervicovaginal recolonization by *Lactobacillus* spp. [11] and could in turn lead to relapse. Indeed, high rates of BV recurrence have already been documented after oral treatment with metronidazole [123]. Moreover, antibiotic therapy is related with side effects [124], lack of efficiency due to resistant strains [125], and interference with the normal vaginal microbiota [126]. In this context, the emergence of novel therapeutic strategies can be helpful to improve treatment outcomes.

Probiotics are “live microorganisms which, when administered in adequate amounts, confer a health benefit on the host” [127]. Therefore, the application of beneficial microbes such as *Lactobacillus* spp. could overcome the presence of pathogens and promote a healthy state to the vaginal microbiome [128]. Over the past years, different evidence has been published acknowledging the clinical relevance of probiotics as an adjuvant component together with antibiotic therapy (Table 1). For example, the consumption of an oral probiotic containing *L. fermentum* 57A, *L. plantarum* 57B, and *L. gasseri* 57C, together with metronidazole, was able to lengthen the intervals without BV when compared to antimicrobial therapy alone [129]. Moreover, administration of vaginal tablets with *L. rhamnosus* BMX 54 as an adjuvant to antimicrobial therapy was shown to improve the rates of healthy vaginal microbiota in BV-diagnosed patients comparing to those receiving antibiotics alone [130]. In agreement to that, oral capsules of *L. rhamnosus* GR-1 and *L. reuteri* RC-14 together with tinidazole significantly reduced Shannon’s diversity of vaginal microbiota in BV-diagnosed women and increased the relative abundance of *Lactobacillus* spp. when compared to patients treated only with antibiotics [131].

The potential of probiotics as a single therapy has also been documented (Table 1). In this context, vaginal suppositories containing *L. rhamnosus* IMC 501 and *L. paracasei* IMC 502 were tested in apparently healthy women. The authors registered 40% of patients with an intermediate Nugent score before the treatment and half of them returned to a normal state (low Nugent score) after the intervention [132]. Besides, when compared to placebo, vaginal capsules of *L. fermentum* 57A, *L. plantarum* 57B and *L. gasseri* 57C were able to significantly reduce vaginal pH and Nugent score, suggesting a shift to a healthy vaginal microbiota profile. Also, probiotic administration caused a transient increase in the abundance of *Lactobacillus* spp. originally present in the capsules. That is, the preparation significantly increased *Lactobacillus* spp. amounts after seven days of treatment, but the levels slowly declined during the following eight days without intervention [133]. Likewise, in vitro co-culture assays showed an antimicrobial activity of *L. acidophilus* GLA-14 and *L. rhamnosus* HN001 against pathogens associated with BV (*Gardnerella vaginalis* and *Atopobium vaginae*) [125]. Altogether, these data suggest a relevant role for *Lactobacillus*-based probiotics concerning the maintenance of vaginal microbiome homeostasis.

Interestingly, *Lactobacillus* spp. strains have been isolated from healthy women and their potential as probiotics has been investigated. *L. plantarum* and *L. fermentum* strains isolated from healthy Cuban women exhibited antagonic activity against *G. vaginalis*, *C. albicans*, as well as remarkable adhesive capacities and lactic acid production [134]. Similarly, *L. fermentum* 9LB6, 4LB16, and 10LB1 and *L. plantarum* 9LB4 strains were isolated from Algerian women and suggested as probiotic candidates based on their relevant inhibitory activity of pathogens, lack of hemolytic capability and other beneficial properties according to in vitro studies [135]. However, the reports mentioned above lack clinical evidence. Based on in vitro assays, *L. crispatus* LbV 88, *L. gasseri* LbV 150N, *L. jensenii* LbV 116, and *L. rhamnosus* LbV96 strains, originally obtained from healthy pregnant women, were selected as relevant for vaginal health [136] and further used in a pilot clinical trial in which a yoghurt preparation containing those beneficial microbes were administered to BV-diagnosed women together with metronidazole. The study showed that the group receiving probiotics significantly improved the recovery rate from BV when compared to patients treated only with antibiotics [137].

Evidence linking probiotics to HPV clearance has been published [11]. When the SiHA cell line naturally infected with HPV-16 was co-cultured with *Bifidobacterium adolescentis*, a reduction in HPV E6 and E7 mRNA production was noticed. Acknowledging the relevance of HPV for cervical cancer development, probiotics could also be linked to cancer prevention. In this context, in vitro studies demonstrated that the combination of *L. gasseri* and *L. crispatus* present cytotoxic effect in HeLa cervical cancer cells naturally infected with HPV-18 and this effect did not occur in normal cervical cell lines [138,139]. An interventional study that recruited HPV-positive women with LSIL reported that patients treated with oral *L. casei* had a significantly higher chance of resolving cytological abnormalities compared to untreated women, yet both groups did not differ significantly with respect to HPV clearance [140] (Table 1). Similarly, a cohort of women with BV or HPV infection and Pap smear abnormalities were exposed to metronidazole/fluconazole along with *Lactobacillus rhamnosus* BMX 54 delivered as vaginal tablets. The authors showed that the group receiving long-term probiotic treatment (for 6 months) exhibited not only a significant higher chance of resolving cytological abnormalities but also showed increased rates of HPV clearance when compared to the group receiving short-term probiotic therapy (for 3 months) [141] (Table 1). Taken together, those reports suggest an interface between the microbiome constitution, HPV clearance and cervical lesions, raising the possibility that microbiota modification can avoid cervical cancer development, yet further clinical trials are required to elucidate this relationship.

Given the potential use of probiotics in vaginal health, new approaches for their administration are being developed. A mucoadhesive vaginal tablet containing *Lactobacillus* spp. was designed with two layers: one for rapid dissolution and fast release and another for prolonged release of remaining microbes [142]. Interestingly, the use of intravaginal probiotics could be further indicated in situations in which a faster restoration of vaginal health is required such as after chemotherapy, radiotherapy, or antibiotics use, while oral probiotics would be recommended to avoid recurrent infections, for example [133]. Moreover, the use of probiotics containing transgenic bacteria has also been proposed. A vaginal tablet of *L. jensenii* 1153–1666 genetically altered in order to express an HIV-1 entry inhibitor (cyanovirin-N protein) promoted a successful vaginal colonization by the strain when tested in macaques [143].

After the currently recognized effectiveness of fecal microbiota transplant, vaginal microbiota transplant (VMT) has also been proposed as an interesting field of research for future probiotic therapy [144]. In this context, the vaginal microbial transfer or vaginal seeding practice for babies delivered by Cesarean section (C-section) has been discussed [145]. The natural transfer of commensal bacteria from mother to infant during vaginal delivery birth is essential for a healthy infant, since the bacteria transferred during this process stimulate the immune system, regulate gut development and produce vitamins for the host [146,147,148]. Several studies have reported an association between C-section and an increase in frequency of asthma, immune disorders, and risk of obesity in the child [149,150,151,152]. Vaginal seeding is a technique in which the vaginal microbiota is transferred from the mother using a cotton gauze or cotton swab with vaginal fluids to the mouth, nose and skin of her newborn [153]. The first study that described vaginal seeding and the microbiota vaginal colonization in newborns was published in 2016 [153]. This study showed that the microbiome of infants delivered by C-section that were exposed to maternal vaginal fluids at birth resembled the microbiota of vaginally delivered infants. However, the health effect on infant’s life promoted by vaginal microbiome transferred artificially via vaginal seeding is still unclear.

In addition to mother-to-infant transplantation, VMT has been recently evaluated as a treatment option for patients with recurrent BV after being subjected to different antimicrobial regimens [154]. Vaginal fluid was collected from healthy donors to test its potential therapeutic effects in five patients with relapsing BV. After being evaluated according to pH and microscopy, the discharge from healthy donors was diluted with 1 mL of sterile saline and introduced in the posterior fornix of the recipients. Out of five patients enrolled, four showed BV long-term remission after receiving the VMT, while the remaining participant experienced a partial response. VMT was able to change the recipient’s microbiome composition. One month after treatment, four out of five patients exhibited a remarkable shift in its microbiome composition. The post-VMT microbiota was characterized by an increase in *Lactobacillus* species coupled with a simultaneous decrease in members of *Bifidobacterium*, *Prevotella,* and other genera. Interestingly, three patients required repeated VMT (and even a donor change for one of them) to reach BV remission [154], suggesting that the VMT dosage and possible donor-recipient specifications are still to be determined with confidence.

A potential risk limiting VMT feasibility is the transmission of pathogens between donor and recipient. DeLong and collaborators proposed a screening protocol for cervicovaginal fluid from *Lactobacillus*-dominated donors in order to reduce the risk of pathogen transmission. The authors suggested a list of tests to perform on donor candidates before VMT, such as HIV (1 and 2), HAV, HBV, HCV, EBV, *Treponema pallidum*, *Chlamydia trachomatis*, *Neisseria gonorrhoeae*, *Toxoplasma gondii,* and others. According to their criteria, from 20 candidates enrolled in the study, only seven were eligible for vaginal fluid donations, indicating the importance of donor screening prior to VMT [155].

## 8. Conclusions

The cervicovaginal microbiome is a dynamic network of microorganisms able to modulate a host’s immune responses and promote an environment susceptible to viral infection acquisition and development of CIN. Recent studies showed the association between high-diversity cervical microbiota and HPV infection, CIN and cervical cancer. On the other hand, women with dominant *Lactobacillus* species (except for *L. iners*-dominant) can promote HPV clearance and the absence of cervical lesions. Thus, specific bacteria or the high diversity microbiota may function as biomarkers for cervical alterations, and can as well as be used to identify women at high risk to develop persistent HPV infection, CIN, and cancer. However, the mechanisms involved in the role of the microbiota on the promotion of, or protection to those conditions are yet to be fully elucidated.

The use of probiotics is demonstrated in vivo and in vitro for HPV clearance and significant CIN regression. Therefore, the manipulation of the microbiota by the use of probiotics or by VMT may be a feasible option to induce HPV infection clearance, CIN regression, and stop progression to cervical cancer.

## Figures and Tables

**Figure 1 ijms-21-00222-f001:**
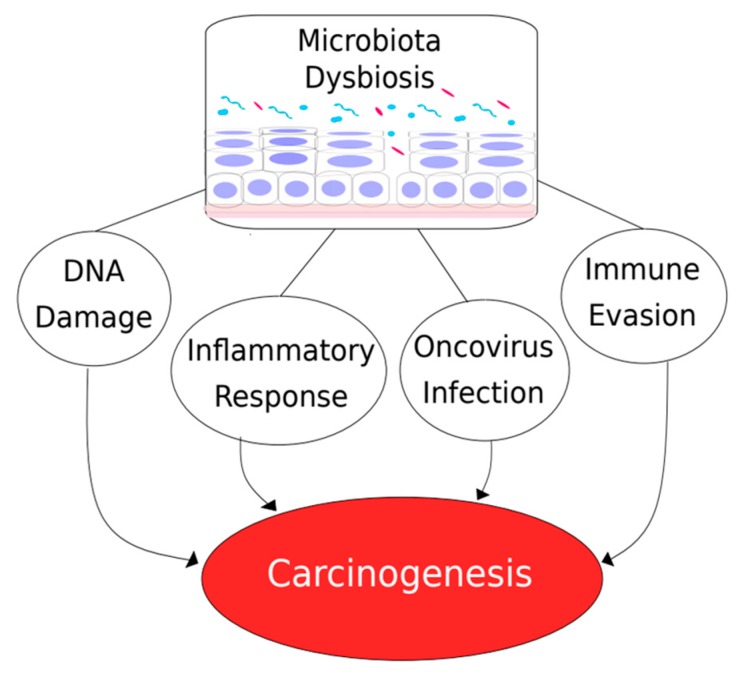
The impact of microbiota dysbiosis in carcinogenesis. Dysbiotic microbiota (blue and pink rods and circles) may drive carcinogenesis either by modulation of host mechanisms, such as promoting immune response alterations and DNA damage, or by directly eliciting tissue damage, thus facilitating infection by oncoviruses.

**Figure 2 ijms-21-00222-f002:**
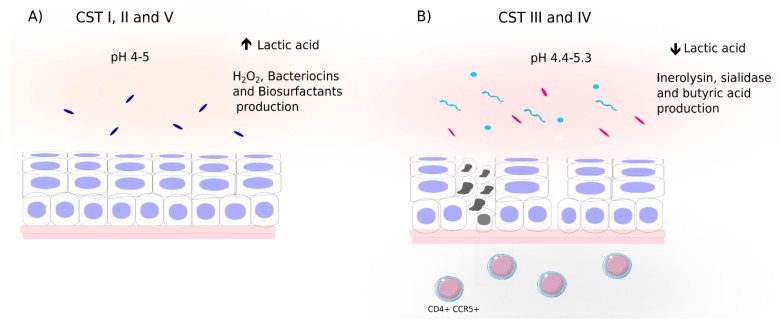
Bacterial bioproducts modulate cervical microenvironment. Bacterial cervical communities named CSTs (community state types) are displayed on the top of the Figure (**A**,**B**) in roman numerals (I, II, III, IV). (**A**) CST I, II, and V contain dominant *Lactobacillus* (non-*iners*) species (dark blue rods) and produce high level of lactic acid, hydrogen peroxide, and bacterial bioproducts (bacteriocins and biosurfactants). (**B**) CST III shows the *Lactobacillus iners*-dominant community (pink rods) and CST IV displays bacterial high diversity (light blue rods and circles) with increased frequency of anaerobic species. They both produce less lactic acid and exhibit inerolysin, sialidase, and butyric acid production. In addition, CST III and IV modulate immune responses by induction of proinflammatory cytokine production and recruitment of CD4+CCR5+ lymphocytes to the cervical region.

**Figure 3 ijms-21-00222-f003:**
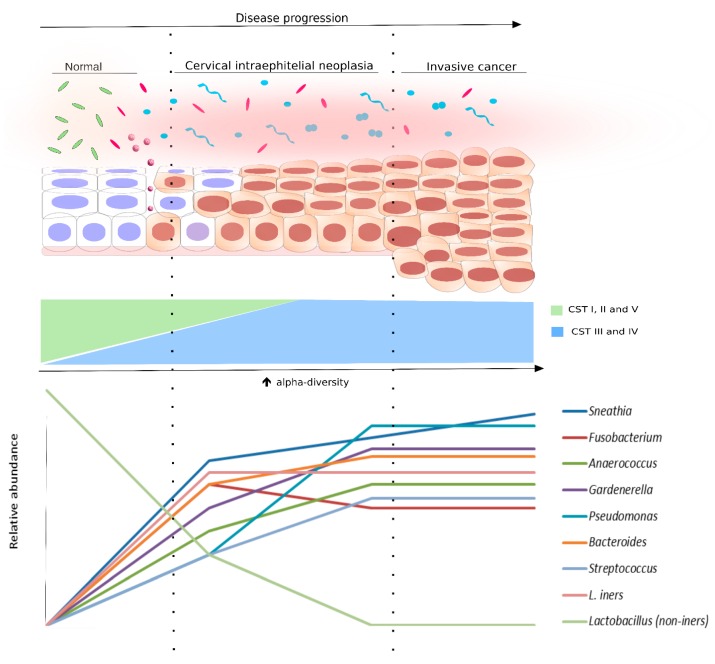
Bacterial diversity distribution in intraepithelial neoplasia progression. The scheme displays the progression of the cervical epithelium from normal to invasive cervical cancer, as well as the bacterial diversity (alpha-diversity) and the species abundance in the cervical microenvironment at each cytological stage. The normal cytology is commonly associated with CSTs I, II, or V, which are *Lactobacillus* species (non-*iners*)-dominant (light green rods). However, following the cervical disease progression, the relative abundance of *Lactobacillus* non-*iners* species start to decrease. Concomitant to that, alpha-diversity increases and the microbiota is changed to CST III (pink rods and circles) or IV (light blue and pink shapes). Some bacterial species were found, in different studies, associated with cervical disease progression. They are also displayed in this figure in a representative graph of relative abundance (lower panel).

**Table 1 ijms-21-00222-t001:** Highlights of clinical trials that explored the use of probiotics to bacterial vaginosis treatment, HPV infection and abnormal cervical cytology. BV, bacterial vaginosis; AV, aerobic vaginitis; VVC, vulvovaginal candidiasis.

Study	Treatment	Study Characteristics	Main Outcomes
Heczko et al., 2015 [129]	Oral metronidazole 500 mg twice daily for seven days together with an oral probiotic preparation (prOVag^®^) containing *L. fermentum* 57A, *L. plantarum* 57B and *L. gasseri* 57C twice daily for ten days.	578 participants (118 receiving antibiotic together with prOVag and 241 treated with antibiotic together with placebo); Patients recruited from nine private gynaecological clinics in Poland; Women with history of recurrent BV/AV and with current symptoms were recruited and underwent five assessments.	Treatment with probiotics lengthened the time to a clinical relapse. The average time to a BV/AV relapse event was 71.4 days for women treated with prOVag and 47.3 days for the placebo group (*p* = 0.0125). Microbiologically confirmed BV/AV patients at visit V were significantly less frequent in the prOVag group (*p* = 0.04632). Nugent score achieved statistically significant differences between visits I and III, I and IV and IV and V for prOVag group. For placebo subjects, differences were found only between visits I and III, III and IV.
Recine et al., 2016 [130]	Oral metronidazole 500 mg twice a day for seven days together with vaginal tablets of *L. rhamnosus* BMX 54 (NORMOGIN^®^). Administration of the probiotic occured once a day for 10 days, twice a week for 15 days and once every 5 days for 7 months.	250 participants (Group A: 125 women subjected to metronidazol alone and Group B: 125 patients receiving antibiotic together with probiotic). Women sexually active, non-pregnant and with BV diagnostic were recruited at University of Rome. Patients were assessed after 2, 6 and 9 months.	After 2 months of treatment, 90.4% of Group B patients showed BV clinical remission, compared to 79.4% in Group A subjects (*p* = 0.014). After 6 months, physiological vaginal microbiota was found in 74.6% of Group B participants, compared to 25.4% of Group A women (*p* < 0.0001). After 9 months, healthy microbiota were observed in 79.7% of Group B subjects, compared to 20.3% in Group A (*p* < 0.001). Vaginal pH was significantly higher in Group A compared to that of Group B at 6-month (*p* = 0.034) and at 9-month (*p* < 0.001) follow-ups.
Laue et al., 2018 [137]	500 mg of oral metronidazole twice a day for seven days together with 125g yoghurt drink twice daily for 4 weeks. The yoghurt drink (*verum*) contained *L. crispatus* LbV 88, *L. gasseri* LbV 150N, *L. jensenii* LbV 116 and *L. rhamnosus* LbV 96	36 participants were randomly assigned to a metronidazole plus probiotic arm (*n* = 18) or a metronidazole plus placebo arm (*n* = 18). Women newly diagnosed with BV were recruited from Schleswig-Holstein region in Germany.	Post-intervention, all women receiving antibiotic plus probiotics showed recovery from BV, while 35.3% of patients after antibiotic plus placebo remained with the condition according to Amsel criteria (*p* = 0.018). Amsel score decreased by 3.41 ± 0.71 for the probiotic group compared to 1.94 ± 1.95 for placebo subjects (*p* = 0.037). Nugent score decreased by 4.65 ± 2.85 for probiotic subjects compared to 2.82 ± 3.59 for the placebo group (*p* = 0.158).
Verdenelli et al., 2016 [132]	Vaginal suppository SYNBIO^®^*gin* containing *L. rhamnosus* IMC 501 and *L. paracasei* IMC 502 once daily for seven days.	35 apparently healthy women from Italy were enrolled. Assessments were made three times: before treatment, immediately after treatment and 21 days after treatment.	After treatment, 50% of the women with an intermediate Nugent score reverted to the normal state. There were no significant differences in vaginal pH comparing the time points before and after the treatment. After SYNBIO*gin*, *L. rhamnosus* IMC 501 and *L. paracasei* IMC 502 exhibited increased abundance in the vaginal microbiota that slowly declined over the following 21 days.
Tomusiak et al., 2015 [133]	InVag^®^ vaginal capsules containing *L. fermentum* 57A, *L. plantarum* 57B and *L. gasseri* 57C once a day for seven days.	160 women of European descent and with dysbiotic vaginal microbiome were enrolled. Patients were randomly assigned either to a group receiving the InVag preparation or to a placebo group. Four visits were included in the trial. Assessments were made at visits I, III and IV.	For InVag subjects, there was a significant reduction in vaginal pH between visits I and III (*p* < 0.0016) and visits I and IV (*p* < 0.0001). For placebo patients, differences were not significant. For the InVag arm, Nugent score decreased significantly from visits I to III (*p* = 0.0001), I to IV (*p* < 0.0001) and III to IV (*p* = 0.0238). For the placebo arm, Nugent score also decreased significantly from visits I to III (*p* < 0.0001) and I to IV (*p* = 0.0002). However, visits III and IV were not significantly different from each other. InVag subjects significantly increased *L. plantarum* and *L. fermentum* in their vaginal microbiota by approximately 1,000 times at visit III and the levels slowly declined until visit IV. In placebo subjects, *L. plantarum* and *L. fermentum* increased much more slowly, by approximately 10 times at visit IV. *L. fermentum* 57A, *L. plantarum* 57B and *L. gasseri* 57C were confirmed to be present on the vaginal epithelium of 82% of InVag participants at visit III and on 47.5% at visit IV.
Palma et al., 2018 [141]	500 mg of metronidazole twice a day for 7 days or daily fluconazole (150 mg) for two consecutive days together with vaginal tablets of *L. rhamnosus* BMX 54 for 3 months (short-term) or 6 months (long-term).	117 subjects were randomly assigned to the short-term probiotic administration (group 1, *n* = 60) or to the long-term *Lactobacilli* implementation (group 2, *n* = 57) at University of Rome. Women diagnosed with yeast vaginitis / BV together with HPV infection / cytological abnormalities were enrolled. Assessments were made before treatment, and at 3, 6 and 9 months after intervention.	3 months after treatment, statistically significant differences were not found between groups 1 and 2. After 9 months, 79.4% of patients subjected to the long-term probiotic administration solved the cytological abnormalities, against 37.5% in group 1 (*p* = 0.041). After 9 months, 11.6% women from group 1 cleared HPV infection, compared to 31.2% from group 2 (*p* = 0.044).
Verhoeven et al., 2012 [140]	Daily consumption of a commercially available probiotic drink (Yakult) containing *L. casei* Shirota during the study period (6 months).	54 HPV+ women with LSIL were assigned to a group receiving probiotics or to a group without intervention (control). The study was developed at the University of Antwerp, Belgium. Assessments were made at study entry (t_1_), 3 months after (t_2_) and 6 months after (end of the study, t_3_).	60% of probiotic-consuming patients solved the cytological abnormalities against 30.7% patients without intervention (*p* = 0.047). After 3 months, 16% of probiotic subjects cleared the HPV infection against 7.7% in control women (*p* = 0.13). After 6 months, 29.2% of probiotic intakers cleared the HPV infection compared to 19.2% of control subjects (*p* = 0.41).

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
