# Peer review of "The Role of the Cervicovaginal Microbiome on the Genesis and as a Biomarker of Premalignant Cervical Intraepithelial Neoplasia and Invasive Cervical Cancer"

_ijms, 2019, doi:10.3390/ijms21010222_

Round 1
Reviewer 1 Report
Figure 3 does not show any "impact" of cervical microbiota on CIN progression but the bacterial diversity distribution in different CIN stages.
Author Response
Figure 3 does not show any "impact" of cervical microbiota on CIN progression but the bacterial diversity distribution in different CIN stages.
We thank the Reviewer for pointing out such inconsistency. As per the Reviewer´s suggestion, we have changed the legend to Figure 3, and described it as "Bacterial diversity composition in...".
We have also noticed the "Moderate English changes required" option marked by the Reviewer, and in response we conducted a full English style revision of the manuscript, correcting several typos and other mistakes that were present in the original submission.
Reviewer 2 Report
The work by Curty et al is an interesting and thorough review outlining the impact of the cervicovaginal microbiome on different aspects of cervical cancer through a detailed analysis of the data reported in the scientific literature. The manuscript is extensively cited and overall well-written. However, some areas of the review remain a bit cumbersome and could benefit from added clarity. To improve its readability, I suggest a small reorganization, which however is to be taken in a non-literal but only suggestive sense.
Chapter 2 (History of bacterial identification): this section is very didactic, with appreciable historical notes, which make reading pleasant. However, the last paragraph (lines 23-29) should be more detailed and the most recent platforms employed in NGS analyses (e.g. Illumina) should be cited.
Chapter 3 (Microbiome and Cancer): lines 17-20 discuss the prevalence of Fusobacterium in cervical cancer. In my opinion it is better to move this paragraph below, for example at line 22, after the introduction of the topic “gynecological cancers”.
Chapter 4 (cervicovaginal microbiota and viral infections): this chapter is very long. It would be useful to facilitate reading, to introduce a subparagraph entitled “Cervicovaginal microbiota”. Consequently, lines 18-29 should be moved after the detailed description of the organization of cervicovaginal microbiota provided at lines 10-41.
Author Response
The work by Curty et al is an interesting and thorough review outlining the impact of the cervicovaginal microbiome on different aspects of cervical cancer through a detailed analysis of the data reported in the scientific literature. The manuscript is extensively cited and overall well-written. However, some areas of the review remain a bit cumbersome and could benefit from added clarity. To improve its readability, I suggest a small reorganization, which however is to be taken in a non-literal but only suggestive sense.
We are delighted that the Reviewer found our manuscript throughly and well written. We have accomodated the suggestions made by the Reviewer to the revised manuscript as detailed below.
Chapter 2 (History of bacterial identification): this section is very didactic, with appreciable historical notes, which make reading pleasant. However, the last paragraph (lines 23-29) should be more detailed and the most recent platforms employed in NGS analyses (e.g. Illumina) should be cited.
As per the Reviewer´s request, we have included additional text and references on more recent NGS platforms that are being used for microbiome analysis. Additions have been highlighted in yellow in the revised manuscript.
Chapter 3 (Microbiome and Cancer): lines 17-20 discuss the prevalence of Fusobacterium in cervical cancer. In my opinion it is better to move this paragraph below, for example at line 22, after the introduction of the topic “gynecological cancers”.
We have made the changes exactly as suggested by the Reviewer in the revised text.
Chapter 4 (cervicovaginal microbiota and viral infections): this chapter is very long. It would be useful to facilitate reading, to introduce a subparagraph entitled “Cervicovaginal microbiota”. Consequently, lines 18-29 should be moved after the detailed description of the organization of cervicovaginal microbiota provided at lines 10-41.
As suggested, two subtopics were created on Chapter 4, "Cervicovaginal bacterial composition and profiles" and "Cervicovaginal microbiota and relationship to viral infections". Text has been moved exactly according to the Reviewer´s suggestions (highlighted in yellow in the revised manuscript).